# A WRKY Transcription Factor CbWRKY27 Negatively Regulates Salt Tolerance in *Catalpa bungei*

Jiaojiao Gu [1,2,†], Fenni Lv [2,†], Lulu Gao [2], Shengji Jiang [2], Qing Wang [2], Sumei Li [2], Rutong Yang [2], Ya Li [2], Shaofeng Li [3,*] and Peng Wang [1,2,*]

1   College of Forestry, Nanjing Forestry University, Nanjing 210037, China
2   Jiangsu Key Laboratory for the Research and Utilization of Plant Resources, Institute of Botany, Jiangsu Province & Chinese Academy of Sciences, Nanjing 210014, China
3   State Key Laboratory of Tree Genetics and Breeding, Experimental Center of Forestry in North China, National Permanent Scientific Research Base for Warm Temperate Zone Forestry of Jiulong Mountain in Beijing, Chinese Academy of Forestry, Beijing 100091, China
*   Correspondence: lisf@caf.ac.cn (S.L.); wp280018@163.com or wp280018@cnbg.net (P.W.); Tel.: +86-010-69826131 (S.L.); +86-025-84347151 (P.W.)
†   These authors contributed equally to this work.

**Abstract:** *Catalpa bungei* is an economically important tree with high-quality wood, which is highly ornamentally valuable in China. Salinity is one of the major constraints restricting the growth of the *C. bungei*. However, the molecular mechanism underlying the salt stress response remains unknown in *C. bungei*. In our previous study, a novel WRKY transcription factor gene *CbWRKY27* was isolated using association mapping based on the transcriptome database of Catalpa Yuqiu1. In this study, *CbWRKY27* was found to function as a transcriptional activator in the nucleus. The transcription of *CbWRKY27* was inhibited under salt stress and reactive oxygen species (ROS) but was induced after abscisic acid (ABA) treatment. CbWRKY27-overexpression plants showed decreased tolerance to salt stress compared to wild type while enhancing sensitivity to ABA-regulated lateral root length. Quantitative real-time PCR (qPCR) studies showed that the transcript levels of the ABA biosynthesis gene (*NCED3*), signaling genes (*ABI3* and *ABI5*), and responsive genes (*RD29B* and *RD22*) were greatly increased in *CbWRKY27*-overexpression plants under salt stress. Under salt treatment, *CbWRKY27*-overexpression plants disturbed ROS homeostasis by repressing antioxidant enzymes and enhancing the production of $O_2^-$ and $H_2O_2$ through down-regulation of ROS-scavenging-related genes (*APX*, *SOD*, and *PER57*). In summary, these results indicate that *CbWRKY27* negatively regulates salt tolerance in *C. bungei*.

**Keywords:** transcription factor; salt tolerance; genetic transformation; plant hormone





## 1. Introduction

There are more than 1.5 million hectares of coastal tidal flats and 100 million hectares of inland saline land in China (Ministry of Natural Resources of the People's Republic of China, 2021). Under salt stress, plant physiology and metabolism are disturbed; the absorption balance of essential nutrient ions is destroyed; and the growth and development of plants are restricted. High salinity reduces the plant growth rate, evaporation rate, pigment synthesis, survival rate, and yield [1,2]. The effects of salt stress include osmotic stress, ionic toxicity, nutritional deficiencies, etc. [3]. Compared with comprehensive soil control, the most cost-efficient tactics is to breed and utilize salt-tolerant cultivars [4]. Hence, understanding the molecular regulatory network involved in salt tolerances is a prerequisite for molecular breeding.

Abscisic acid (ABA), reactive oxygen species (ROS), and other endogenous signaling molecules are significantly altered in plants under salt stress [5]. The plant hormone ABA plays a decisive role in salt stress response. Salt stress induces ABA biosynthesis

while preventing ABA degradation [6]. Previous research has demonstrated that *NINE-CIS-EPOXYCAROTENOID DIOXYGENASE 3* (*NCED3*) is essential for the biosynthesis of ABA during salt stress in *Arabidopsis thaliana* and *Glycine max* [7,8]. Both ABA-dependent and ABA-independent mechanisms govern the induction of *NCED3* in response to salt stress [9]. ROS refers to a general term for a class of oxygen-containing substances with active chemical characteristics and strong oxidative ability, including superoxide radicals ($O_2^-$) and hydrogen peroxide ($H_2O_2$) [10]. Under salt stress, plants accumulate excessive ROS, which cause plant DNA, protein, and lipid damage, damaging the membrane system and causing the plant to lose normal physiological functions [11]. To eliminate the excessive ROS accumulation caused by salt stress, plants have evolved a complex set of defense mechanisms, including enzymatic and non-enzymatic systems [12,13]. The enzymatic system includes superoxide dismutase (SOD), peroxidase (POD), and catalase (CAT) which scavenge free radicals. SOD can catalyze the decomposition of $O_2^-$ into $H_2O_2$ and $O_2$ and then, through POD and CAT enzymatic degradation, generate $H_2O$ and $O_2$ [14]. A previous study indicated that ROS are integral components of ABA signaling networks [15]. In *Arabidopsis*, ABA stimulates the generation of ROS in guard cells, and the double mutant *AtrbohD/F* impairs ABA–induced $H_2O_2$ production, stomatal closing, and ABA-mediated restriction of root growth [16,17]. However, the relationship between ABA and ROS in regulation of salt stress responses is still unclear.

WRKY transcription factors (TFs) are among the largest transcription factor families in plants and are a part of the control of plant development and growth., establishment of stress signal transduction pathways, and expression of related genes in the process of hormone metabolism regulation [18]. WRKY TFs are considered master regulators of molecular reprogramming to enhance stress tolerance in plants [19,20]. Overexpression of *Zea mays ZmWRKY33* enhances salt tolerance in *Arabidopsis* [21–23]. Overexpression of the WRKY TF *GmWRKY12* enhances drought and salt tolerance in Glycine max [24]. Overexpression of *Dendronthema grandiform WRKY5* enhanced tolerance to salt stress in chrysanthemum [25]. Overexpression of *PbWRKY40* enhanced tolerance to salt stress in *Pyrus betulaefolia* [26]. In addition, overexpression of other WRKY genes also reduces salt tolerance in plants. Overexpression of *PalWRKY77* reduces salt tolerance in *Populus* [27]. Heterologous expression of *GmWRKY13* and *Capsicum annuum CaWRKY27* significantly reduces seedling survival under salt stress in *Arabidopsis* [28,29]. Recent studies have revealed that WRKY transcription factors participate in the ROS signaling pathway to adjust plant responses to salt stress [30]. The overexpression of *Gossypium hirsutum GhWRKY39* can improve the POD and SOD activities after salt treatment in cotton [31]. The WRKY transcription factor genes *Oryza sativa OsWRKY30* and *OsWRKY72* are also induced by ROS, and overexpression of these two genes improves rice tolerance to salt stress [32,33].

*Catalpa ungee*, belonging to the genus *Catalpa*, is a valuable garden ornamental, is tolerant to saline conditions, and is a high-quality wood resource tree species in China. However, little is known about the molecular mechanism underlying the salt stress response [34–36]. This lack of knowledge has led to many unsuccessful attempts to enhance *C. bungei*'s salt tolerance. In our previous study, a novel WRKY transcription factor gene, *CbNN1* (new name *CbWRKY27*, homologous to *Arabidopsis WRKY22*), was isolated using association mapping based on the transcriptome database of Catalpa Yuqiu1 [37]. The expression of the gene *CbWRKY27* increased with increasing rooting ability, which indicated that it might play a positive role in adventitious root formation [37]. However, the accurate function of *CbWRKY27* has not been determined. Here, we investigated the expression profile of *CbWRKY27* and its physiological phenotypes in overexpressing transgenic *C. bungei* lines. Transgenic *C. bungei* lines of *CbWRKY27* showed decreased salt tolerance as opposed to the wild type. Furthermore, the expression levels of several genes regulated in reply to ABA and ROS metabolism and signaling were investigated under salt treatment in *C. bungei*. These findings can contribute to the improvement of salt-tolerant *C. bungei* varieties.

## 2. Materials and Methods

### 2.1. Plant Materials, Growth Conditions, and Stress Treatments

Two accessions, NJQ301, wild relatives of *C. bungei* were collected from the Catalpa resource nursery of the Institute of Botany, Jiangsu Province, and Chinese Academy of Sciences, Nanjing, China (32°06′ N, 118°84′ E). Embryogenic calli from the accession NJQ301 were utilized for plant transformation [38]. For stress treatment, six-week-old seedlings were grown in Driver and Kuniyuki Walnut medium (DKW) including 300 mM NaCl, 100 μM ABA, 5 μM SA, and 100 μM $H_2O_2$ for different times (0, 0.5, 1, 2, 4, 8, 12, and 24 h) [39].

For ABA stress, six-week-old seedlings were raised in DKW medium with 0 or 25 μM ABA for three weeks. For salt stress and ABA stress, detached leaves of seedlings were cultured in DKW medium with 0 mmol/L NaCl, 300 mmol/L NaCl, and 300 mmol/L NaCl + 25 mmol/L ABA at different times (24 and 48 h).

For salinity stress in culture room, six-week-old seedlings were raised in DKW medium with 0 or 300 mM NaCl for 48 h for histochemical staining. For salinity stress in an artificial climate chamber, six-week-old seedlings were grown in pots containing vermiculite. After four weeks of growth, seedlings were watered every seven days with 30 mL of 300 mM NaCl solution for different times (10, 15, 20, 25, and 30 days). The control plants were watered with 30 mL water every seven days. The experiment consisted of three biological replicates.

For salinity stress in the greenhouse, the ten-week-old seedlings were grown in pots containing vermiculite. After eight weeks of growth, the plants were watered every day with 300 mL of 200 mM NaCl solution for two days. A quantity of 300 mL of water was used to irrigate the control plants water every 1 day. The experiment consisted of five biology replicates.

Sterile seedlings and embryogenic calli were raised in a culture room at a temperature of 23–25 °C, light intensity of 1500 lx, and under a 14-h light/10-h dark photoperiod. *C. bungei* plantlets and tobacco (*Nicotiana benthamiana*) plants were cultured in an artificial climate chamber at a temperature of 25–28 °C, light intensity of 6000 lx, and under a 14-h light/10-h dark photoperiod.

### 2.2. Subcellular Localization

The full-length coding sequence (CDS) of CbWRKY27 (without a stop codon) was amplified by PCR with enzyme (Vazyme, P505-d1, Nanjing, China) and primers (CbNN1-qRT-F/R). Then compatible fragments were combined into pBinGFP4 between the Kpn I and BamH I restriction sites by homologous recombination reaction to obtain the CbWRKY27-eGFP construct. The primers used are listed in Supplementary Table S1. The recombined vector was transformed into *Agrobacterium tumefaciens* strain EHA105 and transiently expressed in the epidermal cells of tobacco leaves through agroinfiltration [40], whereas an empty vector was utilized as a control. Following a 48-h dark culture, the transformed tobacco leaves were observed under a confocal laser scanning microscope (LSM900, Zeiss, Germany). For 4′,6-diamidino-2-phenylindole (DAPI) staining, the tobacco leaves were infiltrated in DAPI solution (10 μg/mL, Solarbio, D8200, Beijing, China) for 15–30 min to mark nuclei.

### 2.3. Transcription Activation Assay in Yeast

The full-length CDS and different truncated segments of *CbWRKY27* were amplified and fused with the GAL4-DNA-binding domain in pGBKT7 between the EcoR I and Pst I restriction sites. The primers used are listed in Supplementary Table S1. The constructs were transformed into yeast strain AH109 chemically competent cell by the lithium acetate method, according to the manufacturer's instructions (Coolaber, CC300, Beijing, China). Transformed clones were selected on synthetic dropout medium deficient in tryptophan (SD/-Trp, Coolaber, PM2252, Beijing, China). The transcriptional activation activity of *CbWRKY27* in yeast was assayed on SD/-Trp-His (Coolaber, PM2282, Beijing, China) medium supplemented with X-α-D-Galactosidase (X-α-gal, Coolaber, CX11922, Beijing,

China) at 29 °C cultured for 72 h, based on its ability to grow in the absence of His and the blue color reaction.

### 2.4. Yeast One-Hybrid Assay

Yeast one-hybrid assay (Y1H) was used to dissect the binding activity of *CbWRKY27* with the W-box element. The full-length CDS of *CbWRKY27* was cloned into the pB42AD vector at the EcoR I restriction site to generate the *CbWRKY27*-AD construct. An 18-bp oligonucleotide sequence including three tandem repeat copies of the W-box element (5′- TTGACT -3′) was cloned into the Xho I restriction site of the pLacZi reporter vector to generate the W-box-LacZ construct. The vectors mW-box1-LacZ-mW-box4-LacZ were obtained by changing the core TGAC sequence of the W-box elements to TTGAtT, TcGACT, TTaACT, and TTGgCT. The *CbWRKY27*-AD and (m)W-box-LacZ vectors were co-transformed into the yeast strain EGY48 chemically competent cell by the lithium acetate method, according to the manufacturer's instructions (Coolaber, CC302, Beijing, China). The DNA–protein interaction was evaluated founded on blue color development of transformants growing on SD-Trp/-Ura (Coolaber, PM2262, Beijing, China) plates with X-gal (Coolaber, CX11921, Beijing, China) at 29 °C cultured for 72 h. The primers used are listed in Supplementary Table S1.

### 2.5. Vector Construction and Plant Transformation

The full-length CDS of *CbWRKY27* was cloned into the binary vector pCAMBIA2300, driven by the *Cauliflower mosaic virus* (CaMV) 35S promoter, between the Sal I and Xba I restriction sites to create the pCAMBIA2300-*CbWRKY27* plant overexpression vector. Employing our efficient *Agrobacterium*-mediated genetic transformation system, as described before [40], the pCAMBIA2300-*CbWRKY27* vector was transferred into the embryogenic calli of NJQ301. All transgenic lines were confirmed using PCR (*GUS* gene amplification) and qPCR (*CbWRKY27* RNA levels). The primers used are listed in Supplementary Table S1.

### 2.6. Chlorophyll and Carotenoid Determination

Fresh leaves of six-week-old seedlings were collected to measure chlorophyll and carotenoid contents, as previously stated [41]. Briefly, the samples were extracted with 10 mL 80% acetone in the dark (24 h). The absorbance of the extracts was measured at 646, 663, and 470 nm using an ultraviolet spectrophotometer (MAPADA P5, Shanghai, China) to calculate carotenoid, chlorophyll a and chlorophyll b content. Three biological replicates were used in the experiment.

### 2.7. Leaf Gas Exchange Measurement

The net photosynthetic rate, transpiration rate, and stomatal conductance were determined by a portable photosynthetic system (Li6800, LI-COR, Inc. Lincoln, NE, USA) in the morning between 8:00 and 11:00. The $CO_2$ concentration was maintained at 400 μmol·mol$^{-1}$. The light intensity was controlled at 1500 μmol m$^{-2}$ s$^{-1}$, and the relative humidity ranged from 55% to 75%. The temperature inside the leaf chamber was maintained at 25 °C. Five biological replicates were used in the experiment.

### 2.8. Chlorophyll Fluorescence

The maximal photochemical efficiency of PS II in the dark was measured using a chlorophyll fluorometer (OS1p, Boston, MA, USA) in the morning between 9:00 and 11:00. Five to six leaves were selected for measurement and dark-treated for 15 min before the measurement. Five biological replicates were used in the experiment.

### 2.9. Histochemical Staining

Nitroblue tetrazolium (NBT) and 3, 3-diaminobenzidine (DAB) staining were performed to detect ROS. For NBT staining, the leaves were submerged in NBT solution (Jiancheng Bio, I023-1-1, Nanjing, China) until a dark blue color appeared. For DAB stain-

ing, all the samples were submerged in DAB solution (Jiancheng Bio, I026-1-1, Nanjing, China) for 2 h. All samples were then cleared of chlorophyll by using 95% (*v/v*) ethanol. After chlorophyll was completely removed, the residual liquid was washed with distilled water and photographed for storage.

### 2.10. Measurement of ROS Content, Antioxidant Enzyme Activity, and Malondialdehyde (MDA) Content

The $H_2O_2$ content (Jiancheng Bio, A064-1-1, Nanjing, China), $O_2^-$ content (Jiancheng Bio, A052-1-1, Nanjing, China), SOD activity (Jiancheng Bio, A001-4-1, Nanjing, China), POD activity (Jiancheng Bio, A084-3-1, Nanjing, China), and MDA activity (Jiancheng Bio, A003-3-1, Nanjing, China) in the leaves were measured using reagent kits. The leaves of the seedlings were collected with an ice pack and stored at $-80$ °C. Previous to the measurement, the samples were ground with liquid nitrogen, and a phosphate buffer (pH = 7.0) was added, and the mixture was centrifuged at $10,000 \times g$ for 10 min at 4 °C, and the experiment was carried out according to the manufacturer's instructions. The final reaction product was placed in a quartz cuvette, and the absorbance was measured using a spectrophotometer (Mipuda P5, Shanghai, China). Finally, the content was calculated according to the calculation formula. Five biological replicates were used in the experiment.

### 2.11. Quantitative Real-Time PCR Analysis

Total RNA was extracted using a Vazyme kit (Vazyme, RC401, Nanjing, China), and cDNA was synthesized using a cDNA kit from Genesand (Genesand biotech, SR512, Beijing, China). Amplification of *CbActin* was utilized to normalize the amount of gene-specific qPCR product. Primers were designed using Primer premier 5 (www.premierbiosoft.com/primerdesign, accessed on 24 July 2022; Supplementary Table S1) and synthesized by SIPUJIN (Nanjing, China). The qPCR was carried out using SYBR Premix Ex Taq II (Takara, RR820, Beijing, China) in the Applied Biosystems Step One Plus TM Real-Time PCR System (Applied Biosystems, Waltham, MA, USA) according to the manufacturer's instructions. The following program was used for qPCR: 95 °C for 30 s, 95 °C for 5 s followed by 40 cycles of 60 °C for 30 s, 95 °C for 15 s, and 60 °C for 1 min. Three biological and technical replicates were performed for every sample. Relative gene expression levels were analyzed using the $2^{-\triangle\triangle CT}$ method [42].

The deduced protein sequences of five ABA related genes (*AtNCED3*, *AtABI3*, *AtABI5*, *AtRD22*, *AtRD29B*) and four ROS homeostasis-related genes (*AtAPX*, *AtSOD*, *AtPER57*, *AtRBOHA*) were downloaded from the National Center for Biotechnology Information (NCBI) database. The BLASTP algorithm was used to search the *C. bungei* genome with a cut-off e-value of $10^{-10}$. All the candidate genes were confirmed by the Pfam database (http://pfam.xfam.org/, accessed on 24 July 2022).

### 2.12. Statistical Analysis

Data were analyzed using one way analysis of variance ANOVA. Comparisons were performed using least significant difference (LSD) tests. All statistical analyses were performed using the two-sample equal variance t-test, and statistical significance was set at $p < 0.05$ and 0.01. GraphPad Prism software version 8.0 (San Diego, CA, USA) was used to present the results graphically.

## 3. Results

### 3.1. CbWRKY27 Is a Nuclear Protein with Transcriptional Activation Activity and W-Box DNA Binding Activity

A previous study predicted that *CbWRKY27* was located in the nucleus [43]. To confirm the subcellular localization of *CbWRKY27*, fusion plasmid of *CbWRKY27*-eGFP and eGFP alone were transiently transformed into the young leaves of tobacco plants. The confocal microscopy images indicated that the *CbWRKY27*-eGFP fusion protein was exclusively localized in the nuclei of tobacco leaf epidermal cells (Figure 1A). As a control, the nucleus and cytoplasm both contained the eGFP protein (Figure 1A).

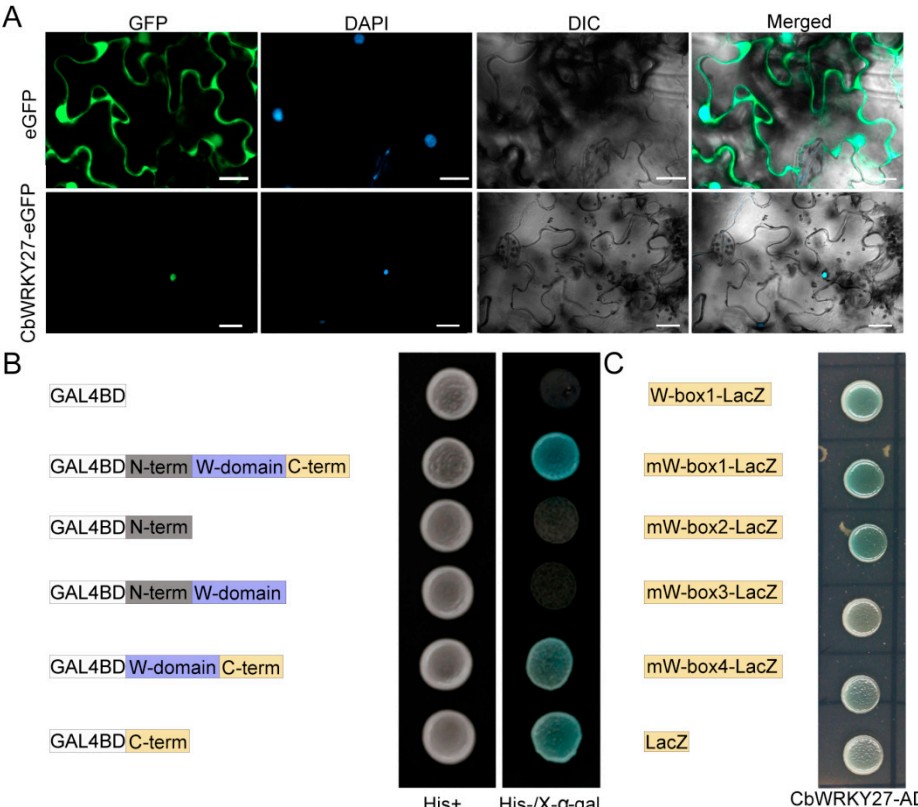

**Figure 1.** *CbWRKY27* functions as a transcription factor. (**A**) Subcellular localization of *CbWRKY27*-eGFP in tobacco (*Nicotiana benthamiana*). 35S::eGFP and 35S::*CbWRKY27*-eGFP were transiently expressed in tobacco leaves via agroinfiltration. The nuclei of *N. tabacum* epidermal cells were stained with 4′,6-diamidino-2-phenylindole (DAPI). Bar = 20 μm. (**B**) The transcriptional activation activity assay of *CbWRKY27* in yeast cells. The C terminus of *CbWRKY27* significantly promoted the growth of yeast cells, which was accompanied by a blue tone on selection media with X-gal. (**C**) Y1H assays to determine the binding activity of *CbWRKY27* to W-box sequences. W-box and mutant W-boxes (mW-box1-mW-box4) were fused to the upstream region of the LacZ reporter gene. The empty pLacZi was used as a negative control.

To research whether *CbWRKY27* possessed transcriptional activation activity, we fused *CbWRKY27* or its truncated fragments with the GAL4 DNA- binding domain of the pGBKT7 vector. The yeast cells transformed with pGBKT7-*CbWRKY27* (1-179 aa), pGBKT7-*CBWRKY27* (1-236 aa) and the empty vector pGBKT7 only managed to survive on the SD/-Trp medium (Figure 1B). The yeast cells transformed with pGBKT7-*CbWRKY27* (1-352 aa), pGBKT7-*CbWRKY27* (180-352 aa), and pGBKT7-*CbWRKY27* (237-352 aa) grew well on SD/-Trp/-His medium and became blue when exposed to X-α-gal (Figure 1B). These outcomes demonstrated that the W-domain and C-terminal regions of *CbWRKY27* function as transcriptional activator.

To investigate whether *CbWRKY27* can bind to W-box elements presented in the promoter regions of target genes, we performed a yeast one-hybrid (Y1H) assay to assess the direct binding activity between *CbWRKY27* and the W-box core sequence TTGACT. *CbWRKY27* could bind directly to the W-box element even though the TTGACT core sequence was changed to TTGAtT or TcGACT (Figure 1C). However, when the core sequence altered to TTaACT and TTGgCT, it did not make yeast turn blue (Figure 1C). These findings indicate that *CbWRKY27* can directly bind W-box elements in the promoter regions of target genes, and that this binding is specific.

### 3.2. Expression Patterns of CbWRKY27 under Diverse Stress Treatments

To better understand the role of *CbWRKY27*, the transcript abundance of *CbWRKY27* under NaCl, ABA, SA, and $H_2O_2$ treatments was examined using quantitative real-time PCR (qPCR). The expression levels of *CbWRKY27* decreased by 68.09% after 4 h of salt treatment compared to that of the control and then slowly increased after 8 h to 24 h of salt treatment (Figure 2A). In contrast, *CbWRKY27* was rapidly induced by ABA and SA, with increased expression. The RNA of *CbWRKY27* increase to the highest level of 4.63-fold and 3.16-fold after 1 h of treatment and then decreased to a lower level after 4 h to 24 h of ABA and SA treatments, respectively (Figure 2B,C). Interestingly, under $H_2O_2$ treatment, the expression levels of *CbWRKY27* decreased by 66.87% after 4 h treatment but increased to the highest level of 1.72-fold after 12 h of treatment (Figure 2D). The speedy induction of *CbWRKY27* under diverse stress treatments demonstrated its critical function in the response to abiotic stresses in *C. bungei*.

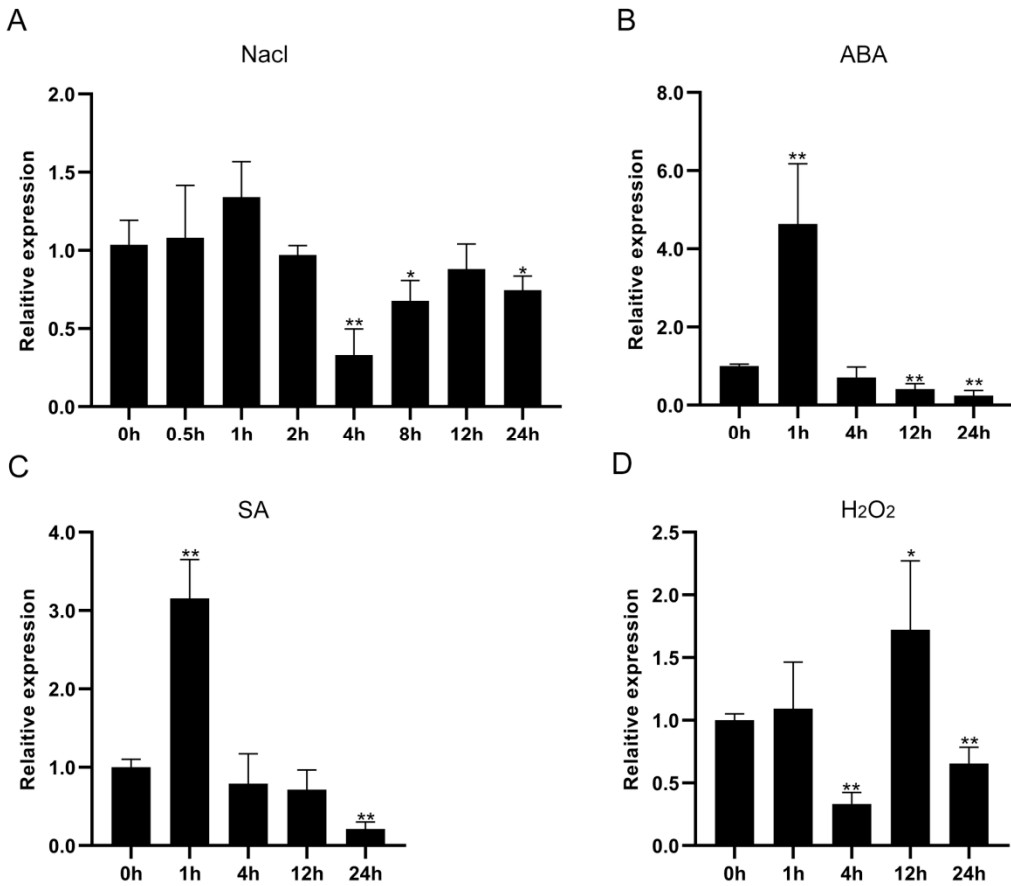

**Figure 2.** Expression levels of *CbWRKY27* under diverse stress treatments. Total RNA extracted from whole plants exposed to (**A**) 300 mM NaCl, (**B**) 100 µM abscisic acid (ABA), (**C**) 5 µM salicylic acid (SA), and (**D**) 100 µM $H_2O_2$ were used for Quantitative real-time PCR (qPCR). The data represent the means ± SD (n = 3), * $p < 0.05$ and ** $p < 0.01$ using Student's *t*-test. The six-week-old seedlings were subjected to the treatment.

### 3.3. Overexpression of CbWRKY27 Decreases Salt Tolerance in C. bungei

To examine the response of *CbWRKY27* to salt stress, we generated overexpression lines of *CbWRKY27* (Supplementary Figure S1) and detected the dynamic phenotypes at 10, 15, 20, 25, and 30 days in 10-week-old seedlings under the 300 mM NaCl treatment. We observed that the leaves of overexpression (OE) lines were more affected than those of wild type at 15 days after NaCl treatment (DANT) (Figure 3A). The survival rate of three OE lines was 87.78%–94.58% whereas that of the wild type remained at 100% at

15 DANT (Figure 3B). Notably, the survival rate of three OE lines dramatically decreased to 5.56%–17.08% meanwhile that of the wild type was 39.17% at 30 DANT (Figure 3B).

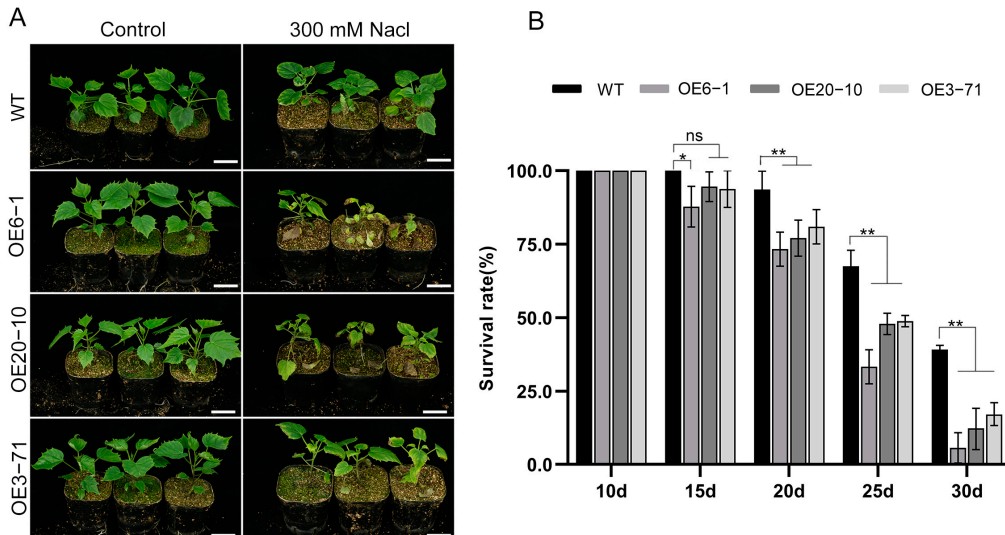

**Figure 3.** Overexpression of *CbWRKY27* reduces survival rate in transgenic plants under salt treatment. (**A**) Phenotypic comparison of wild type (WT) and overexpression plants after 15 days under salt stress (300 mM NaCl). (**B**) Survival rates of overexpression plants and WT at 10, 15, 20, 25, and 30 days after continuous salt stress. Bar = 4 cm. The data represent the means ± SD ($n$ = 5), * $p < 0.05$ and ** $p < 0.01$ using Student's *t*-test.

Wild type (WT) and OE lines that were 18 weeks old were irrigated with 200 mM NaCl for two days in a greenhouse to assess how well adult transgenic plants performed under salt stress. After salt treatment, the OE lines were more damaged than WT (Supplementary Figure S2). Next, we measured net photosynthetic rate, stomatal conductance, and transpiration rate. There were no significant differences in three photosynthetic gas exchange parameters between the OE lines and WT under the control condition (Figure 4A–C). However, photosynthesis of OE plants was lower more than that in WT under salt stress (Figure 4A–C, Supplementary Figure S2). We also contrasted the maximum efficiency of PSII (Fv/Fm) values under NaCl stress between the OE lines and WT. Under control condition, the Fv/Fm values of OE lines and WT remained at approximately 0.8 (Figure 4D). Under NaCl treatment, the *CbWRKY27* OE lines showed lower Fv/Fm values than those of WT (Figure 4D).

### 3.4. Overexpression of CbWRKY27 Enhances Sensitivity to ABA under Salt Stress

To study *CbWRKY27* sensitivity to ABA, six-week-old OE lines and WT with adventitious root length of 1.5 cm were grown on DKW medium supplemented with 0 or 25 µM ABA for 20 days (Supplementary Figure S3). When ABA is not present, the leaf color of OE lines and WT remained green, whereas the leaf color of OE lines changed to white-yellow or white and remained green in the WT under 25 µM ABA conditions (Figure 5A). Under normal conditions, the lateral root length of OE lines and WT was comparable. However, in a medium supplemented with 25 µM ABA, the OE lines showed shorter lateral root length than the WT plants (Figure 5B, Supplementary Table S2). In contrast, the number of lateral roots of OE lines and WT with ABA treatment increased by 40.31% and 31.15%, respectively, compared with that of plants without ABA treatment, indicating that ABA improved the number of lateral roots in *C. bungei* (Figure 5C, Supplementary Table S2). According to these findings, the overexpression of *CbWRKY27* enhances sensitivity to ABA in *C. bungei*.

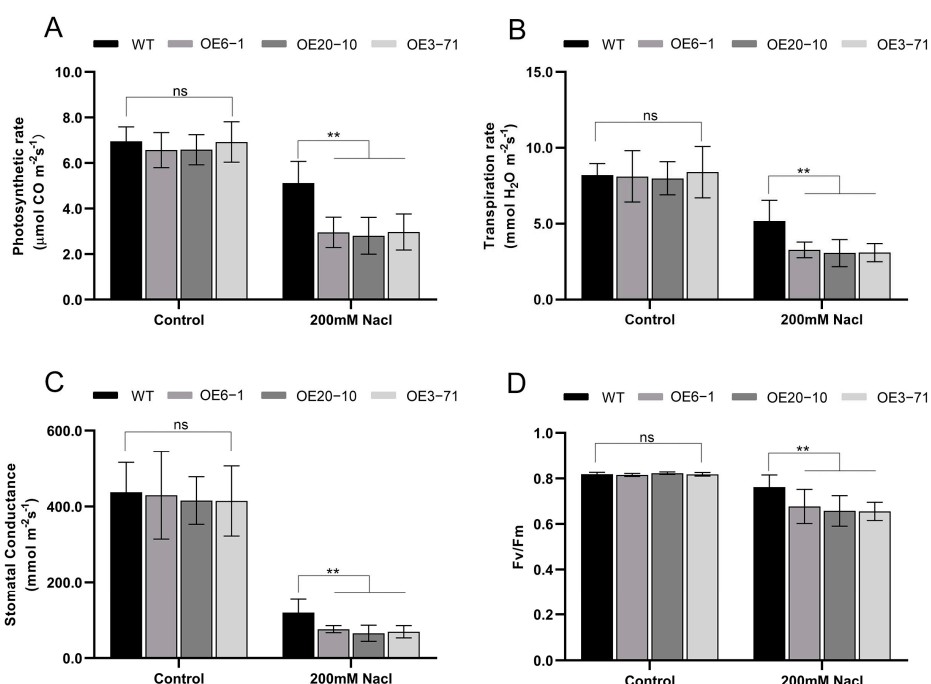

**Figure 4.** Overexpression of *CbWRKY27* reduces photosynthetic performances in transgenic plants under salt treatment. (**A**) Photosynthetic performances of the WT and OE lines with or without salt treatment (200 mM Nacl). Net photosynthetic rate ($\mu$molCO$_2$ m$^{-2}$s$^{-1}$). (**B**) Transpiration rate (mmolH$_2$Om$^{-2}$s$^{-1}$). (**C**) Stomatal conductance (mmolH$_2$Om$^{-2}$s$^{-1}$). (**D**) The maximum efficiency of PSII (Fv/Fm). The data represent the means $\pm$ SD ($n$ = 5), ** $p < 0.01$ using Student's *t*-test.

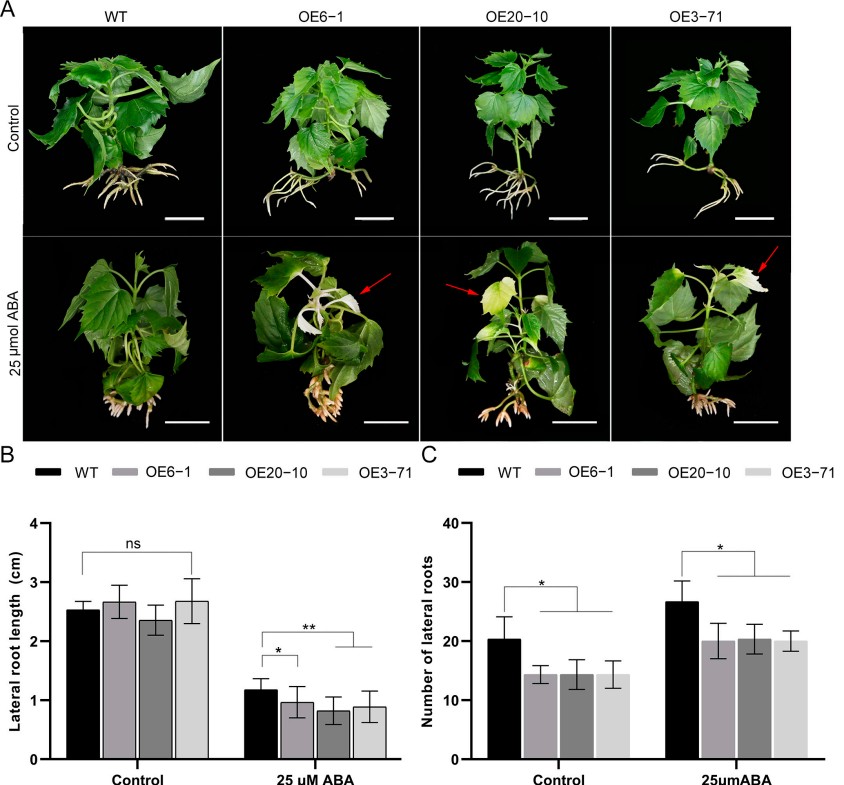

**Figure 5.** Overexpression of *CbWRKY27* enhances ABA sensitivity in transgenic plants. (**A**) Phenotype of OE lines and WT treated with 25 or 0 $\mu$mol ABA. (**B**) Length of lateral roots and (**C**) Number of lateral roots before and after ABA treatment. Bar = 2cm. The red arrow points to the whitened part of the leafThe data represent the means $\pm$ SD ($n$ = 3), * $p < 0.05$ and ** $p < 0.01$ using Student's *t*-test.

To further test the response of *CbWRKY27* to ABA under salt stress, the leaves of the OE lines and WT were treated with DKW medium supplemented with NaCl or ABA for 24 h. In the absence of NaCl, the leaf growth status of the OE lines was similar to that of WT (Figure 6A). However, OE lines had more withered leaves than WT under the 300 mM NaCl treatment. Notably, under 300 mM NaCl + 25 μM ABA treatment, ABA reduced the harm that salt in transgenic plants' leaves produced (Figure 6A). After 300 mM NaCl + 25 μM ABA treatment, the contents of chlorophyll and carotenoid in OE lines were higher than those in salt treatment, yet they are still less than those in WT (Figure 6B–E). These results also showed that ABA is involved in the response of *CbWRKY27* to salt stress in *C. bungei*.

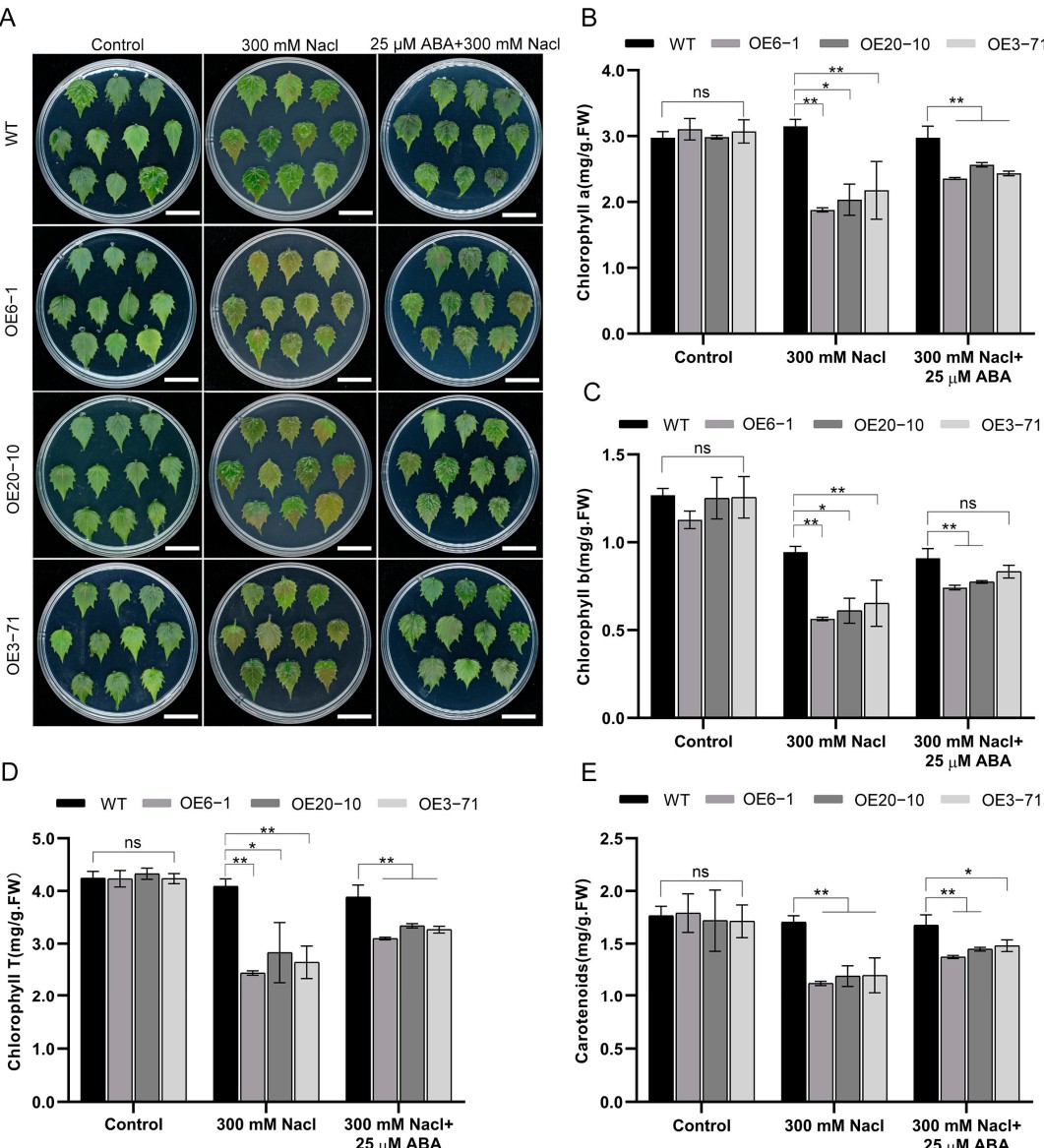

**Figure 6.** Exogenous ABA affects the salt tolerance of CbWRKY27-overexpressing plants. (**A**) Phenotypic analysis of detached leaves of transgenic plants and WT with 0 mM NaCl, 300 mM NaCl, and 300 mM NaCl+ 25mM ABA. Measurement of (**B**) chlorophyll A, (**C**) chlorophyll B, (**D**) total chlorophyll, and (**E**) carotene. Bar = 4 cm. The data represent the means $\pm$ SD ($n$ = 3), * $p$ < 0.05 and ** $p$ < 0.01 using Student's *t*-test. The six-week-old seedlings were subjected to treatment.

To clarify the role of *CbWRKY27* in regulating salt tolerance via an ABA–dependent pathway, qPCR was performed on OE6-1 and WT seedlings treated with 300 mM NaCl. Five ABA-responsive genes, including *NCED3*, *ABI3*, *ABI5*, *RD22*, and *RD29B*, were analyzed.

The transcript levels of all five genes in the OE line were significantly greater than those in the WT after being exposed to salt for 4 h (Figure 7). The transcript levels of *NCED3* in the OE6-1 line increased 5.33-fold compared to those in the WT under normal conditions (Figure 7). Notably, the total RNA of *NCED3* in OE line dramatically increased up to 30.69-fold compared to that in the WT 4 h after salt treatment (Figure 7). These results showed that *CbWRKY27* responds to salt stress by enhancing the expression of ABA-related genes.

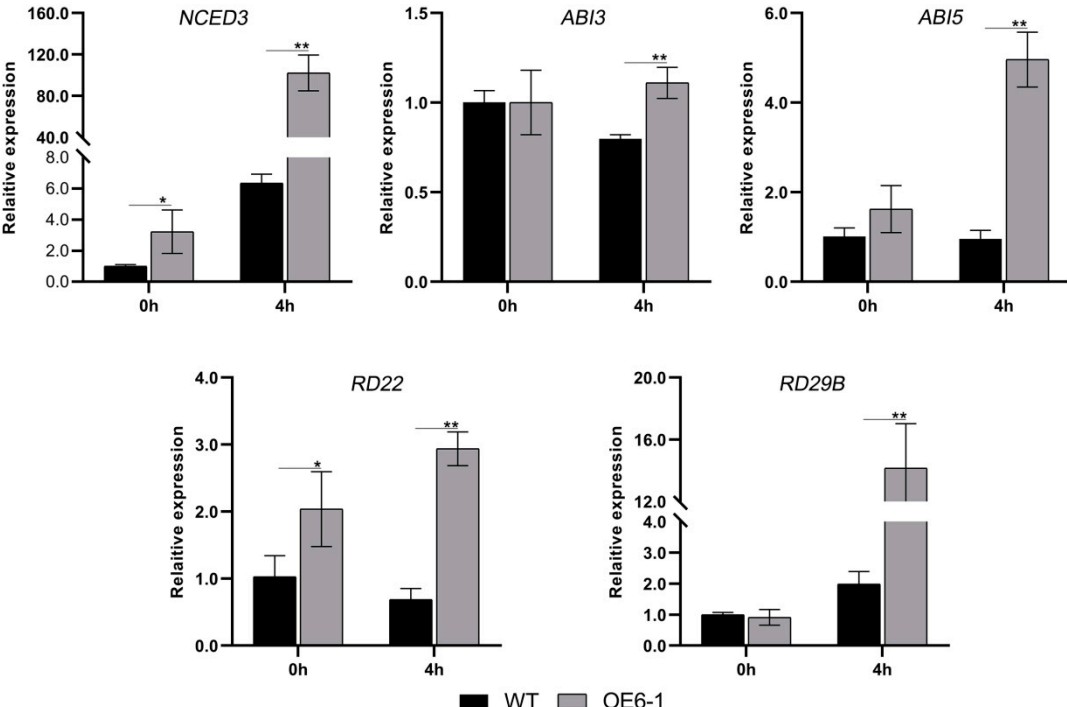

**Figure 7.** Expression patterns of ABA-related genes regulated by CbWRKY27 under salt treatment. 9-cis-epoxycarotenoid dioxygenase 3 (NCED3), abscisic acid insensitive 3 (ABI3), abscisic acid insensitive 5 (ABI5), responsive to dehydration 22 (RD22), and responsive to desiccation 29B (RD29B). qPCR data were normalized to the internal control expression of CbActin. The $2^{-\Delta\Delta Ct}$ method was used to evaluate quantitative variation between replicates. The data represent the means $\pm$ SD (n = 3), * $p < 0.05$ and ** $p < 0.01$ using Student's t-test. The six-week-old seedlings were subjected to 300 mM NaCl treatment.

### 3.5. Overexpression of CbWRKY27 Enhances the Accumulation of ROS under Salt Treatment

To investigate the role of *CbWRKY27* in the response to ROS under salt stress, ROS content was assessed in transgenic lines and WT after NaCl treatment. The finding of the DAB and NBT staining of detached leaves of six-week-old individuals demonstrated that the leaves of each plant in the control under normal circumstances had no visible staining, whereas the DAB and NBT staining of detached leaves of the transgenic lines after 300 mM NaCl treatment was darker in color than the WT (Figure 8A,B). Moreover, the $O_2^-$ and $H_2O_2$ contents in the OE6-1 line increased up to maximum levels of 1.31-fold and 63.50%, respectively, compared to those in the WT after 300 mM NaCl treatment (Figure 8C,D). These findings suggest that *CbWRKY27* overexpression encourages the buildup of ROS, superoxide, and $H_2O_2$ in *C. bungei*.

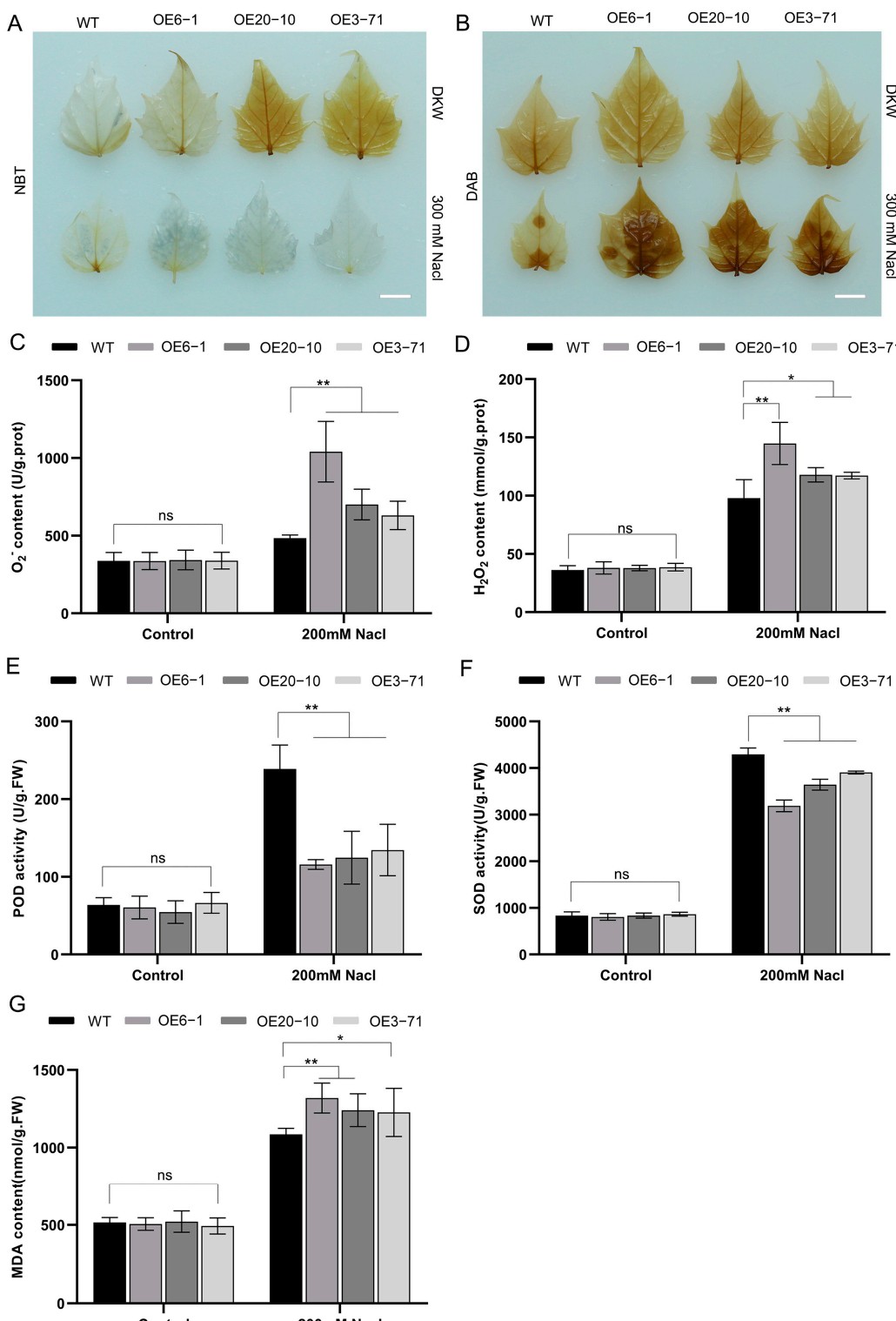

**Figure 8.** Overexpression of *CbWRKY27* enhances the ROS accumulation under salt treatment. The six-week-old seedlings were subjected to 300 mM NaCl treatment (A–B). The 18-week-old seedlings were subjected to 200 mM NaCl treatment (C–G). NBT staining (**A**), DAB staining (**B**), $O_2^-$ contents (**C**), $H_2O_2$ contents (**D**), MDA contents (**E**), POD contents (**F**), and SOD contents (**G**) were measured in the WT and OE lines after salt treatment. Bar = 1cm. The data represent the means ± SD (*n* = 5), * *p* < 0.05 and ** *p* < 0.01 using Student's *t*-test.

The total activities of POD and SOD were significantly decreased by 52.74% and 18.10%, respectively, in 18-week-old OE lines compared with those in WT plants after 200 mM NaCl treatment (Figure 8E,F). Malonaldehyde (MDA) concentration, which indicates membrane damage and membrane lipid peroxidation, served as additional confirmation of these findings. The basal level of MDA in the OE lines was similar to that in the WT under normal conditions (Figure 8G). However, the MDA level in the three OE lines was 17.90%–25.25% higher than that in the WT in response to salt stress treatments. These results indicate that the overexpression of *CbWRKY27* inhibits ROS-scavenging capacity and enhances cell membrane injury under salt stress.

To elucidate the *CbWRKY27* reacts to ROS under salt stress, qPCR was performed on OE6-1 and WT seedlings treated with 300 mM NaCl. Four ROS-related genes, *APX*, *SOD*, *PER57,* and *RBOHA,* were analyzed. Among the four genes, only the transcript levels of *PER57* in the OE6-1 line were significantly lower than those in the WT under normal conditions (Figure 9). The total RNA of *RBOHA* in the OE line increased by 59.81% compared with that in the WT 4h after salt treatment (Figure 9). However, the transcript levels of *APX*, *SOD*, and *PER57* in OE6-1 line decreased by 72.40%, 57.45%, and 50.82%, respectively, compared with those in WT 4h after salt treatment (Figure 9). These findings showed that *CbWRKY27* reacts to salt stress by suppressing the expression of genes linked to ROS scavenging and enhancing the expression of RBOHA.

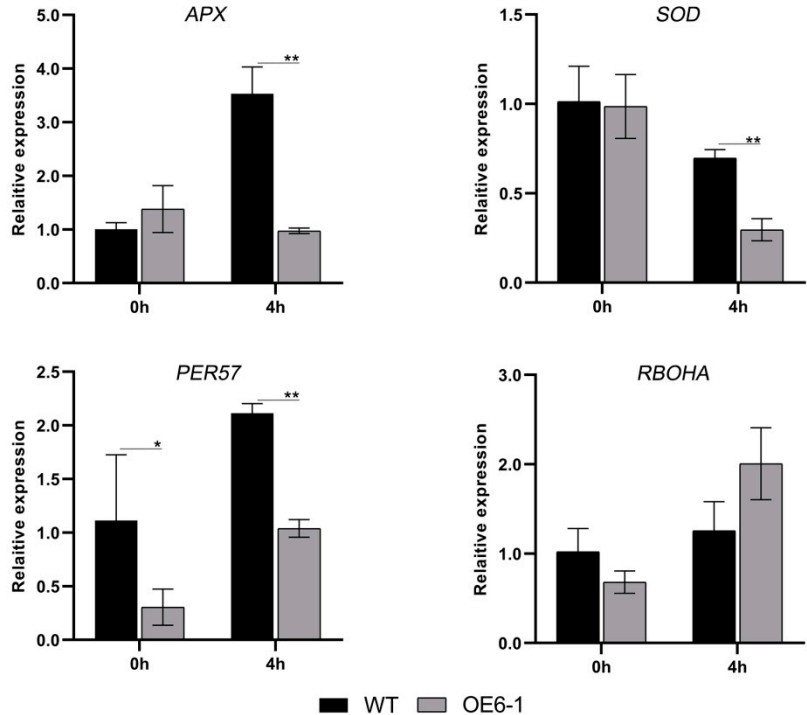

**Figure 9.** Expression patterns of ROS-related genes regulated by *CbWRKY27* under salt treatment. *Ascorbate peroxidase* (*APX*), *SUPEROXIDE DISMUTASE* (*SOD*), *peroxidase 57* (*PER57*), and *respiratory burst oxidase homologue A* (*RBOHA*). The qPCR data were normalized to the internal control expression of *CbActin*. The $2^{-\Delta\Delta Ct}$ method was used to evaluate the quantitative variation between the examined replicates. The data represent the means $\pm$ SD (*n* = 3), * *p* < 0.05 and ** *p* < 0.01 using Student's *t*-test. The six-week-old seedlings were subjected to 300 mM Nacl treatment.

## 4. Discussion

WRKYs are a type of transcription factors that are extensively found in plants, and they have roles in the regulation of a number of physiological processes, such as growth and development, secondary metabolism, hormone signal transduction, and biotic and abiotic stresses [44,45]. Growing evidence in various plants suggests that WRKYs are crucial

in response to abiotic stress [46,47], including *Arabidopsis*, *Oryza sativa*, *Triticum aestivum*, *Nicotiana tabacum*, *Zea mays*, *Pyrus betulaefolia*, and *Populus* [26,27,48–52]. However, little is known about the functional roles and mechanisms of WRKYs in *C. bungei*. Our previous study showed that *CbWRKY27* belongs to subgroup II and has the highest similarity with *AtWRKY22*. This study's research of the biochemical properties of the *CbWRKY27* protein revealed that *CbWRKY27* localizes to the nucleus and maintains transcriptional activation and W-box DNA binding activity in yeast cells (Figure 1). These results imply that *CbWRKY27* may perform its roles in biotic and abiotic stress responses in *C. bungei*.

Previous studies have shown that WRKYs are induced by exposure to various abiotic stressors [53]. In *Arabidopsis*, *AtWRKY22* is induced by $H_2O_2$ and regulates the expression of salicylic acid- and jasmonic acid-related genes in response to biotic stimuli [54]. In *Gossypium hirsutum*, the Group II WRKY gene *GhWRKY17* is significantly activated at the transcriptional level upon exposure to PEG6000, NaCl, or ABA [55]. In this study, *CbWRKY27* was swiftly repressed to the lowest expression levels after 4 h of treatments with NaCl and $H_2O_2$ and induced the greatest expression levels at 4 h after treatments with ABA and SA (Figure 2). Overexpression of *CbWRKY27* in *C. bungei* decreased the survival rate of seedlings under NaCl treatments, indicating that *CbWRKY27* negatively regulated salt tolerance in *C. bungei* (Figure 3). The chlorophyll and carotenoids levels are considered important factors in salt tolerance analysis [56,57]. Overexpression of *CbWRKY27* in *C. bungei* decreased the chlorophyll and carotenoid contents of the detached leaves of seedlings under NaCl treatments, indicating that *CbWRKY27* is a salt-sensitive gene (Figure 6). Similar phenotypes that are sensitive to abiotic stress have also been observed for other WRKY TF genes. For example, *ZmWRKY114* expression is down-regulated by salt stress, while up-regulated by ABA treatments, and overexpression of *ZmWRKY114* in rice reduces height, root length, and survival rates under salt-stress conditions [58].

Plants' physiological changes can be significantly influenced by ABA, a key regulator of the abiotic stress response [59]. Recent studies have found that WRKYs are involved in ABA signaling in response to abiotic stresses in plants. Under different stressors, ABA buildup may limit some physiological processes such lateral root length, seedling growth, and plant development [60]. In the present study, the sensitivity of CbWRKY27-overexpressing lines to exogenous ABA was significantly enhanced, and the lateral root length and number of lateral roots were significantly reduced (Figure 4). Moreover, exogenous ABA reduced the damage caused by salt stress in detached leaves of the CbWRKY27-overexpressing lines (Figure 6). According to certain research, ABA plays a part in the stress response [61]. For example, the *Populus alba* WRKY transcription factor *PaWRKY77* regulates the increased number of lateral roots, root dry weight, and sensitivity of transgenic plants to salt stress by lowering the ABA level [27].

According to earlier investigations, several WRKYs regulate ABA-responsive genes in response to abiotic stress [62,63]. To shed light on the molecular role of CbWRKY27 in the response to salt stress, the expression levels of five well-characterized marker genes in the transgenic line were determined (Figure 7). Previous studies have shown that the expression of two ABA-inducible marker genes, *RD22* and *RD29B*, is induced under salt stress in *Arabidopsis* [64,65]. The qPCR results of gene expression showed that two marker genes, *CbRD22* and *CbRD29B*, were considerably up-regulated in transgenic *C. bungei* plants under salt treatment, indicating that *CbWRKY27* may work in an ABA-dependent pathway. NCEDs mediate ABA biosynthesis, and *NCED3* is stimulated by both ABA and NaCl. The induction of *NCED3* enhances ABA accumulation, which leads to increased stress tolerance in plants [66,67]. Overexpression of *GmWRKY16* in *Arabidopsis* increases *AtNCED3* expression levels, allowing transgenic plants to withstand salt and drought stress [68]. Unexpectedly, *CbNCED3* was determined in *CbWRKY27* transgenic plants at higher levels than in the WT, which indicated that *CbWRKY27* negatively regulates salt tolerance in *C. bungei* (Figures 6 and 7). *ABI3* and *ABI5* genes are the greatest characterized positive regulators of ABA signaling [69]. Overexpression of *GhWRKY6-like* in *Arabidopsis* improves transgenic plants' resistance to salt stress, by increasing the expression of *AtABI5* [63].

However, under salt treatments, two ABA-responsive genes, *CbABI3* and *CbABI5*, were induced in the *CbWRKY27* transgenic plants, resulting in salt sensitivity in *C. bungei* (Figure 7). These findings suggest that *CbWRKY27* negatively regulates plant salt tolerance through the regulation of ABA pathways.

ROS are important in signal transduction and mediate tolerance to various stresses in plants [70]. Low levels of ROS can trigger the stress response, but high levels of ROS buildup in response to salt stress result in oxidative damage to proteins and nucleic acids, induce membrane lipid peroxidation, and reduce PSI and PSII activities [71–73]. In this study, salt treatment increased the $H_2O_2$ and $O_2^-$ contents and decreased SOD and POD activities in *CbWRKY27* transgenic plants (Figure 8). Moreover, under salt treatment, the net photosynthetic rate and the Fv/Fm value in transgenic lines significantly decreased compared to those in the WT (Figure 4). Other WRKY genes yielded comparable outcomes, such as *GhWRKY17*, which enhances the salt sensitive of cotton by increasing ROS accumulation; *GhWRKY6-like*, which improves salt tolerance of cotton by scavenging of reactive oxygen species [71]; and *PcWRKY33*, from *Polygonum cuspidatum*, which reduces salt tolerance in transgenic *Arabidopsis thaliana* by decreasing activities of ROS scavenging enzymes [74]. The MDA content is a sign of the damaging consequences of ROS under stressful situations [75]. In this study, under salt stress, the MDA contents in the OE lines were significantly higher than that in the WT (Figure 8). Plants have developed a variety of antioxidant mechanisms to prevent oxygen damage and decouple the harmful effects of ROS on plants [76]. The overexpression of *P. deltoides PTP1* reduces salinity tolerance in *Populus* by elevating ROS and MDA levels repressed by ROS-scavenging genes [77]. In this study, qPCR results showed that the expression of ROS-scavenging genes (*CbAPX*, *CbSOD*, and *CbPER57*) in *CbWRKY27* transgenic lines was significantly lower than that in WT (Figure 9). These findings imply that CbWRKY27 can reduce the ROS scavenging capacity of transgenic lines in order to increase their salt sensitivity.

ABA and ROS play critical roles in the mediation of abiotic stress responses in plants [78]. It has been reported that ABA can regulate the expression of ROS–scavenging genes [79,80]. Overexpression of *OsMADS25* altered ROS levels by changing ABA sensitivity under salt treatment [81]. In *Arabidopsis*, ABA makes *pmt1* mutants more sensitive to salt by altering the ROS distribution in the root tip [82]. In this study, the expression of *CbWRKY27* was induced by ABA at 1 h and repressed by $H_2O_2$ at 4 h (Figure 2). Corresponding to the alteration in ABA sensitivity, altered ROS accumulation was also observed under salt conditions (Figures 6 and 8). These results strengthen the link between the altered ABA response and changes $H_2O_2$ levels in *CbWRKY27* transgenic plants.

## 5. Conclusions

In summary, the WRKY TF CbWRKY27 was found to be located in the nucleus with transcriptional activation and W-box DNA binding activities. Overexpression of *CbWRKY27* in *C. bungei* significantly increased the sensitivity to salt stress in transgenic plants. Further research revealed that under salt stress, transgenic plants showed higher ABA sensitivity. Additionally, the transgenic plants showed lower enzyme activities of POD and SOD and higher MDA content than the WT under salt stress. The results indicate that *CbWRKY27* may play a negative role in plant adaptation to salinity conditions by mediating ABA response and ROS homeostasis.

**Supplementary Materials:** The following supporting information can be downloaded at: https://www.mdpi.com/article/10.3390/f14030486/s1, Figure S1: Expression analysis of *CbWRKY27* in wild-type and transgenic lines (OE3-1, OE6-1, OE20-10, OE3-71, OE3-14); Figure S2: Phenotypic map of 18-week-old wild-type and transgenic plants after treatment with 200 mM NaCl for two days; Figure S3: Root growth of six-week-old wild-type and transgenic seedlings before ABA treatment; Table S1: Primer sequences; Table S2: Overexpression of *CbWRKY27* enhances ABA sensitivity in transgenic plants.

**Author Contributions:** P.W. and S.L. (Shaofeng Li) performed project conception and experiment design. J.G. and F.L. conducted the experiments and analyzed the data. L.G. and S.J. performed subculturing of calli. Q.W., S.L. (Sumei Li) and R.Y. performed regenerated shoots transplantation. J.G. and F.L. wrote the original manuscript; P.W., Y.L. and S.L. (Shaofeng Li) revise and finalize the manuscript. All authors have read and agreed to the published version of the manuscript.

**Funding:** This research was partly funded by the Chinese Academy of Forestry-Special funds for basic scientific research service expenses of the central level public welfare research institutes, Grant No. CAFYBB2020QD001, partly funded by the National Natural Science Foundation of China, Grant Nos. 32071794, 32101550, 32271917, partly funded by Jiangsu Agricultural Science and Technology Innovation Fund, Grant No. CX(22)3051.

**Data Availability Statement:** All relevant data can be found within the manuscript and its supporting materials.

**Conflicts of Interest:** The authors declare that they have no known competing financial interests or personal relationships that could have appeared to influence the work reported in this paper.

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
