# Peer review of "A WRKY Transcription Factor CbWRKY27 Negatively Regulates Salt Tolerance in Catalpa bungei"

_forests, doi:10.3390/f14030486_

Round 1

Reviewer 1 Report

In the current study, a WRKY transcription factor (CbWRKY27) was genetically analyzed in Catalpa bungei tree to understand its functional role under salt stress. CbWRKY27 was overexpressed in C.bungei and analysed by qPCR assays and antioxidant enzyme activities under stress conditions. The results of the study indicated a negative regulatory role of CbWRKY27 under salinity stress with disruption in ROS hemostasis and repression of antioxidant enzymes.  The experimental design of the study was appropriate and the results were novel. However, it is not possible to be published in the journal due to the deficiencies written below.

1.       English of the article is not suitable for publication. Due to many linguistic errors and not-completed sentences in the text, it is not possible to understand the whole article. The text should be proofread by a native speaker before publication.

2.       The authors mentioned that Catalpa bungei is tolerant to saline conditions but little is known about its response to salinity. How can be a sentence could include contrary situations? The authors should give references indicating the tolerance of the tree to salinity.

3.       Why did you choose CbWRKY27? There are many transcription factors related to salinity. The literature background about the gene should be given.

4.       The material method section is not appropriate. First off all the stress treatments and growth conditions should be explained more clearly. plant transformation protocol should be explained in detail based on previous references.

5.       There are no references for Subcellular localization, Transcription activation assay… etc. How did you do all these experiments without previous knowledge?

6.       Although the tables and pictures are good to represent the results, their interpretations in the result and discussion section are not sufficient.

7.       There is no clear and logical explanation for the negative regulatory role of the CbWRKY27 under salinity stress.

8.       If this transcription factor is negatively regulating the salinity stress response, where we can use it?

Author Response

Dear reviewer,

Thank you so much for your valuable comments and constructive suggestions to improve our manuscript entitled “A WRKY transcription factor CbWRKY27 negatively regulates salt tolerance in Catalpa bungei” (Manuscript ID:forests-2210536 ). According to your suggestions, we have tried our best to revise relevant parts of the manuscript. We hope that you find this revised manuscript satisfactory for publication in Forests.

Reviewer 2 Report

Comment and suggestion for authors:

The manuscripts describe several approaches to analyze “A WRKY transcription factor CbWRKY27 negatively regulates salt tolerance in Catalpa bungei”. The subject is of interest because salt tolerance is a crucial step to achieve successful propagation. However, the quality of the English language must be checked and the following aspects must be addressed by the authors.

Abstract:

Comment: L22: Please mentioned briefly how CbWRKY27 was isolated in the current study.

Comment: L23: Briefly explain the methodology after line 22 before jumping on the results.

 Introduction:

Comment: L88: C. bungei (modify the specie’s name into italic in the whole ms).

Materials and methods:

Comment: L101: As mentioned in the MS only accession NJQ301 has been used in the study then NJQ305 should be removed.

Comment: L112-113: Please mention the growth stage too

Comment: L113: Seven days instead of 7 days

Comment: L117: Please mention the growth stage too

Comment: L119: Two days

Results:

Comment: L119: Figure 1 (A-C)The picture quality is lower suggested to improve the resolution.

Comment: L238-239: Nicotiana benthamiana and fN. tabacum  (italic).

Comment: L241: Please change the CBWRKY27  into italic in the whole ms.

Comment: L248:  Write H2O2 instead of H2O2 and correct the word rep-resent into represent.

Comment: L249:  Please change the structure of the sentence in the whole MS to “ *p < 0.05 and **P < 0.01 using the Student’s t-test.” Instead of  =3), 248 *p < 0.05 by the Student’s t test, **P < 0.01 by the Student’s t test.

Comment: L273 & 276:  use increase instead of rose.

Comment: L316:  Please provide the morphological data of adventitious rooting including the AR number and length

Comment: L317-319:  How could you be sure that the leaf color changes with 25 uM ABA? the change does not seem to  be significant, please explain.

Comment: L344:  Figure 4 (A), increase the resolution as its difficult to observe the significant differences in leaf color

Comment: L357:  why were only two-time points selected for qRT-PCR assay?

Comment: L370-372: Rephrase the sentence it's difficult to get the point.

Comment: L374-376: also rephrase this sentence.

Comment: L379: Space between ROS and superoxide

Discussion:

Comment: L488 and 496: space between “MDAcontent”. & WT(Fig.7). 

Note: Moreover please provide some recent findings in the discussion and introduction section related to WRKY TF CbWRKY27 in abiotic stress in model tree species such as Populus t. etc. Likewise, the concise and easily understandable outcome of the study and its future implications should be explained in the conclusion section

Author Response

(The authors gave the same response as above.)

Reviewer 3 Report

The authors presented an elegant characterization of a WRKY transcription factor from the tree species Catalpa bungei, CbWRKY27, focused on salt stress response. Good experimental procedures were used to show the activity of CbWRKY27 as a transcription factor and how it responds to salt stress. By producing transgenic plants overexpressing the CbWRKY27 gene, the authors were able to characterize the function of this transcription factor during salt stress and its relationship to the ABA hormone. Further experiments could be performed to strengthen the characterization of CbWRKY27 as a transcription factor, such as luciferase assays or EMSA. In the discussion, more in-depth exploration of the results would benefit the authors.

Below, general comments are provided, and in the attached PDF file, specific comments are given to hopefully help the authors improve the manuscript.

General comments

In the Methods, it would be helpful to provide more clear details about stress experiments and their controls. For example, were control samples (no treatment) obtained during the initial stresses of NaCl, ABA, SA, and H2O2 at 0, 0.5, 1, 2, 4, 8, 12, and 24 hours? Evaluating gene expression in time-points requires a control set of samples, since many genes are influenced by the time of the day (circadian rhythm, light intensity, etc.). More detailed information of how the stress experiments were designed would improve the understanding of the manuscript.

In the Results, it would be interesting to have an in silico analysis of the presence of W-boxes in the promoter regions of the genes with altered expression in the overexpressing transgenic lines, to consolidate the hypothesis of WRKY-mediated regulation of these targets. The images are very beautiful, congratulations!

In the Discussion, more in-depth discussions could be made for the first results (transcription activation in yeast, binding to W-boxes, etc.). It is important to recognize that the transcription activation assay in yeast has limitations, and how these limitations could be overcome in future studies. In the case of binding to W-boxes, the binding of CbWRKY27 to mutated W-box sequences could be discussed. Do other WRKY TFs bind to non-canonical sequences? If yes, provide examples. If not, suggest reasons. This would strengthen the presented results.

In the Conclusion, it would be interesting to provide future perspectives, next steps, and possible applications of the characterization of this transcription factor. An ending phrase summarizing how this work contributes to knowledge of tree biology would also be interesting.

Author Response

(The authors gave the same response as above.)

Round 2

Reviewer 1 Report

the manuscript is now ready for publication